# ReinforceWalk: Learning to Walk in Graph with Monte Carlo Tree Search

**Yelong Shen,**[*] **Jianshu Chen,**[*] **Po-Sen Huang,**[*] **Yuqing Guo & Jianfeng Gao**
Microsoft Research, Redmond, WA 98072, USA
{yeshen, jianshuc, pshuang, yuqguo, jfgao}@microsoft.com

## Abstract

We consider the problem of learning to walk over a graph towards a target node for a given input query and a source node (e.g., knowledge graph reasoning). We propose a new method called ReinforceWalk, which consists of a deep recurrent neural network (RNN) and a Monte Carlo Tree Search (MCTS). The RNN encodes the history of observations and map it into the Q-value, the policy and the state value. The MCTS is combined with the RNN policy to generate trajectories with more positive rewards, overcoming the sparse reward problem. Then, the RNN policy is updated in an off-policy manner from these trajectories. ReinforceWalk repeats these steps to learn the policy. At testing stage, the MCTS is also combined with the RNN to predict the target node with higher accuracy. Experiment results show that we are able to learn better policies from less number of rollouts compared to other methods, which are mainly based on policy gradient method.

## 1 Introduction

We consider the problem of learning to walk over a graph $\mathcal{G} = (\mathcal{N}, \mathcal{E})$, where $\mathcal{N}$ is a set of node and $\mathcal{E}$ is a set of edges, in order to find a target node $n_T \in \mathcal{N}$ for a given pair of source node $n_S \in \mathcal{N}$ and query $q$. Such problem appears in, for example, knowledge graph completion (KBC), where $\mathcal{G}$ is a given knowledge graph, the nodes in $\mathcal{N}$ are different entities and the edges in $\mathcal{E}$ are the relations between two connected nodes (see Figure 1). The objective of the KBC task is to predict a tail entity (represented as a target node $n_T$) given a head entity (represented as a source node $n_S$) and a target relation (denoted as the query $q$). For example, in Figure 1, for the given head entity $n_S =$ Obama and the query $q =$ Citizenship, we start from the source node Obama and walk through the graph to find the target node $n_T =$ USA. The problem could be understood as using the graph $\mathcal{G}$, the source node $n_S$ and the query $q$ as the inputs to predict the target node $n_T$; that is, we want to construct a function $f(\mathcal{G}, n_S, q)$ to predict $n_T$. However, the functional form of $f(\cdot)$ is generally unknown and has to be learned from a training dataset, which consists of a collection of samples in the form of $(n_S, q, n_T)$. For this reason, the problem could not be solved by conventional search algorithms such as $A^*$-search, which seeks to find paths between the given source and target nodes. Instead, the search *agent* needs to learn its search policy from the given training dataset so that for an unseen pair of query and source node, it could walk over the graph to find the correct target node. However, since each training sample is in the form of "(source node, query, target node)", there is no intermediate supervision for the correct search path. Instead, it only receives delayed *evaluative* feedbacks: when the agent correctly (incorrectly) predicts the target node, the agent will receive a positive (zero) reward. Therefore, the agent should be trained by reinforcement learning (RL) (Sutton & Barto, 1998) instead of supervised learning. There are two major challenges of the problem: (i) the problem is partially observable as it usually requires the entire history of observations to make a correct decision[1], and (ii) the reward is sparse as it appears at the end of a search path. Interestingly, the RL formulation of this problem has another useful property: the environment transition model

---

[*]Equal contribution.

[1]For example, in the KBC example in Figure 1, having access to the current node $n_t =$ Hawaii alone is not sufficient to know the best action is to move to $n_{t+1} =$ USA. Instead, it requires the agent to track the entire history, including the input query $q =$ Citizenship, to reach this decision. In next section, we will describe how to train the agent to walk through the graph.

is known and deterministic.[2] This important knowledge will be exploited to develop an effective learning and prediction algorithm for our problem, named as ReinforceWalk, which develop below.

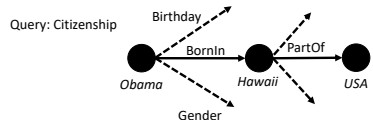

Figure 1: The graph walk in Knowledge Base Completion task

## 2 THE REINFORCEWALK ALGORITHM

Motivated by these observations, we develop a deep neural network architecture for the graph-walking agent, named *ReinforceWalk*. The ReinforceWalk agent consists of a deep structured recurrent neural network (RNN) and a Monte Carlo Tree Search (MCTS). The RNN addresses the partial observability by encoding the history of observations and maps it into the Q-function, the policy and the value function, i.e., it jointly models the Q-network, the policy network and the value network (see Figure 2).We integrate the RNN with MCTS in both training and testing stages, in order to exploit the available model-free and the model-based information. One important contribution of this work is that we combine MCTS with RNN to develop a new reinforcement learning algorithm that effectively learns policy from sparse rewards. By exploiting the knowledge in the environment transition model and the RNN, the MCTS is able to generate trajectories that have significantly more positive rewards than using the policy network alone. From these trajectories, we update and improve the RNN policy in an off-policy manner using Q-learning. This is in sharp contrast to many existing methods for solving the graph walk problem, which use policy gradient method. Policy gradient method requires a large amount of rollouts to obtain a trajectory with positive reward, especially in the early stage of learning. In consequence, our approach is able to learn better policies with less rollouts, as demonstrated by our experiments on several benchmarks, including a synthetic task (in Appendix A) and a real knowledge base completion task. In addition, the ReinforceWalk agent also combines the learned networks with MCTS in testing stage to predict the target node more accurately.

Let $\theta \triangleq \{\theta_S, \theta_A, \theta_\pi, \theta_v, \theta_q\}$ collect all the model parameters to be learned. A popular method to train the agent is the policy gradient method. However, policy gradient method generally has low sample efficiency, especially when the reward signal is sparse, such as in our problem, where the reward ($\{0, +1\}$) appears only at the end of a trajectory. Furthermore, policy gradient usually requires a large amount of Monte Carlo rollouts in order to obtain a trajectory with positive reward, especially in the early stage of learning. To overcome these challenges and by the fact that the state transition model $p(s_{t+1}|s_t, a_t)$ is known, we combine the policy network $\pi_\theta$ with MCTS to generate trajectories with more positive rewards and using these trajectories to further improve the policy $\pi_\theta$. If we are able to learn $\pi_\theta$ from these trajectories, we then further improve $\pi_\theta$. However, since these trajectories are generated by a policy that is different from $\pi_\theta$, they are off-policy data and we could not use policy gradient method to update $\pi_\theta$ due to its on-policy nature. For this reason, we instead update the Q-network from these trajectories using Q-learning, which will automatically update the policy network $\pi_\theta$. Our proposed reinforcement learning algorithm repeatedly applies this policy improvement step. Finally, the value network $V_\theta(s)$ is updated by fitting $V_\theta(s)$ into the terminal rewards $r(s_T, a_T)$. The gradients are calculated by back propagation (through time) over the deep recurrent neural network.

At testing stage, we use the learned policy and value networks with MCTS to generate an MCTS search tree, in a same way as the training stage. Note that each MCTS leaf state $s_T$ is associated with a candidate node $n$ in the graph. However, different MCTS leaf states may correspond to a same node in $\mathcal{G}$ because there could be different paths in $\mathcal{G}$ that lead towards the same node. For this reason, we use the following formula to calculate the score for each *unique* candidate $n$ $\text{Score}(n) = \sum_{s_T \to n} \frac{N(s_T, a_T)}{N} \times V_\theta(s_T)$, where $N(s_T, a_T)$ is the number of times the MCTS edge $(s_T, a_T)$ is visited (stored at MCTS edges), and $N$ is the total number of MCTS simulations. We pick predicted target node $\hat{n}_T$ to be the one with largest score.

---

[2]Briefly speaking, this is because whenever the agent takes an action (by selecting an edge connected to the next node) we will know beforehand which node the environment will transit to.

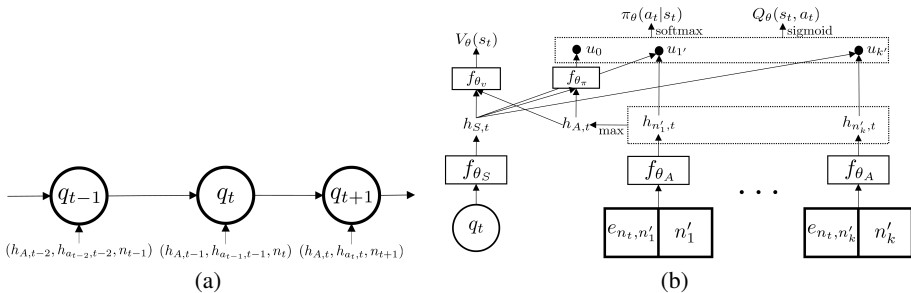

Figure 2: The deep structured RNN architecture for ReinforceWalk. (a) We use an RNN with gated recurrent units (GRU) to encode $s_{t-1}$, $a_{t-1}$ and $n_t$ into a vector $q_t$. (b) We use deep feedforward neural networks to further encode $q_t$ along with other quantities $\mathcal{E}_{n_t}$ and $\mathcal{N}_{n_t}$ into a high-level embedding vectors $\{h_{S,t}, h_{n'_1,t}, \ldots, h_{n'_k,t}, h_{A,t}\}$, which form a hidden representation of the state $s_t$. Finally, they are mapped into the Q-value, the policy and the state value at different output units. Note that the vectors $h_{A,t}$ and $h_{a_t,t}$ will be further fed into the GRU-RNN in (a) to compute the $q_{t+1}$. Therefore, parts (a) and (b) together become a deep structured RNN to model $Q_\theta$, $\pi_\theta$ and $V_\theta$ jointly.

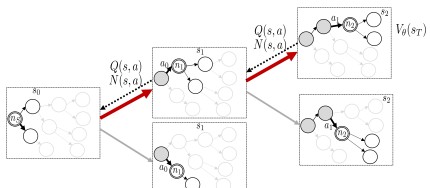

Figure 3: The Monte Carlo Tree Search in ReinforceWalk. The red path is a trajectory generated by MCTS using the PUCT (Rosin, 2011; Silver et al., 2017).

## 3 EXPERIMENTS

We evaluate our approach in a knowledge graph completion task to examine the effectiveness of our approach. Following the setup in Xiong et al. (2017), we use the NELL-995 (Carlson et al., 2010) dataset to evaluate the knowledge graph completion task. The dataset is collected from the 995th iteration of the NELL system. The detailed hyperparameters are described in Appendix B.2.1. In evaluation, the model needs to rank the candidate entities. We use the mean average precision (MAP) scores for each query relation as our evaluation metric. To have more comprehensive comparison, we report the final test MAP scores of ReinforceWalk (RW) on the knowledge base task (for all the 10 relations) in Table 1, and compare it to RL-based methods such as policy gradient (PG), advantage actor-critic (A2C), MINERVA (Das et al., 2017), and DeepPath (Xiong et al., 2017) as well as non-RL baselines such as PRA (Lao et al., 2011), TransE (Bordes et al., 2013) and TransR (Lin et al., 2015).

Table 1: NELL-995 Link Prediction Performance Comparison using MAP scores.

| Tasks | RW | PG | A2C | MINERVA[a] | DeepPath | PRA | TransE | TransR |
|---|---|---|---|---|---|---|---|---|
| athletePlaysForTeam | **0.831** | 0.769 | 0.700 | 0.630 | 0.750 | 0.547 | 0.627 | 0.673 |
| athletePlaysInLeague | **0.974** | 0.955 | 0.955 | 0.837 | 0.960 | 0.841 | 0.773 | 0.912 |
| athleteHomeStadium | **0.905** | 0.865 | 0.861 | 0.557 | 0.890 | 0.859 | 0.718 | 0.722 |
| athletePlaysSport | **0.985** | 0.962 | 0.971 | 0.916 | 0.957 | 0.474 | 0.876 | 0.963 |
| teamPlaySports | **0.881** | 0.631 | 0.679 | 0.751 | 0.738 | 0.791 | 0.761 | 0.814 |
| orgHeadquaterCity | 0.943 | 0.935 | 0.928 | 0.947 | 0.790 | 0.811 | 0.620 | 0.657 |
| worksFor | **0.786** | 0.758 | 0.758 | 0.752 | 0.711 | 0.681 | 0.677 | 0.692 |
| bornLocation | **0.786** | 0.767 | 0.766 | 0.782 | 0.757 | 0.668 | 0.712 | 0.812 |
| personLeadsOrg | **0.821** | 0.802 | 0.810 | 0.771 | 0.795 | 0.700 | 0.751 | 0.772 |
| orgHiredPerson | 0.843 | 0.832 | 0.839 | 0.860 | 0.742 | 0.599 | 0.719 | 0.737 |
| Overall | **0.876** | 0.828 | 0.827 | 0.780 | 0.809 | 0.697 | 0.723 | 0.775 |

[a] We retrain the MINERVA model to follow the setting in Xiong et al. (2017), where each relation is trained separately.

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

# A  ADDITIONAL EXPERIMENTS

## A.1  DESCRIPTION OF TASKS

We briefly describe the two tasks used to evaluate our method. The detailed experimental setup and hyperparameters are described in Appendix B.

**Three Glass Puzzle**  We first study the proposed ReinforceWalk algorithm in a synthetic three glass puzzle (Ore, 1990) dataset. This is a problem studied in math puzzles and graph theory (Ore, 1990). Specifically, there are three milk containers, denoted as containers $A$, $B$ and $C$, respectively, and their respective capacities are $A$, $B$, and $C$ liters. None of the containers has markings to measure the amount of remaining liquid except its total capacity. At each time step, we take one of the three feasible actions on a container: (i) *fill* it (to its capacity), (ii) *empty* (dump) all its liquid, and (iii) *pour* its liquid into another container. The objective of the problem is, for a given target volume, say, $q$ liters, we take a sequence of actions on the three containers so that at least one of them has $q$ liters. Using the graph walk formulation in Section 1, each node $n \in \mathcal{N}$ represents the amounts of liquid remaining in containers $A$, $B$ and $C$, respectively, and each edge represents one of the three feasible actions that could be taken. Furthermore, the desired volume $q$ is the input query. The task succeeds if any one of the containers reaches the desired volume $q$. The training examples are in the form of $(a_S, b_S, c_S, q)$, where $a_S$, $b_S$ and $c_S$ denotes the starting volumes of the three containers. Figure 4 gives a concrete example, where an action sequence is shown to reach the desired volume of $4$. We generate 600 unique three glass puzzle as the synthetic dataset, where 500 samples are used for training and the rest 100 samples are used for testing. In each sample, the agent starts from the initial state and takes an action to move to the next state, until the agent reaches the maximum number of actions (15 in this experiments) or the agent takes the action "STOP". If the final state contains the target volume, the reward is one and zero otherwise. We use policy gradient (PG) and Advantage Actor-Critic (A2C) (Wang et al., 2016) as the baselines. We use the task success rate as the evaluation metric.

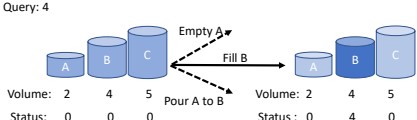

Figure 4: Graph traversal in Three Glass Puzzle problem

**Knowledge Graph Link Prediction**  In knowledge graph link prediction task, the goal is to find the target relation given the initial entity and query relation. Following the setup in Xiong et al. (2017), we use the NELL-995 (Carlson et al., 2010) dataset to evaluate the knowledge graph completion task. The dataset is collected from the 995th iteration of the NELL system. NELL-995 dataset contains 154,213 triples with 75,492 unique entities and 200 unique relations. For each triple $(h, r, t)$, we append $(t, r^{-1}, h)$ to the dataset to connect tail entity $t$ and head entity $h$ with the reverse relation $r^{-1}$. We study the 10 relation task in Xiong et al. (2017) independently. For each relation $r_i$ task, we remove all triples with $r_i$ or $r_i^{-1}$ from the knowledge graph. We split the removed triples into training and testing samples. We use previous proposed algorithms: TransE (Bordes et al., 2013), TransR (Lin et al., 2015), PRA (Lao et al., 2011), and DeepPath (Xiong et al., 2017) as baselines. The detailed hyperparameters are described in Appendix B.2.1. In evaluation, the model needs to rank the candidate entities. We use the mean average precision (MAP) scores for each query relation as our evaluation metric.

## A.2  PERFORMANCE OF REINFORCEWALK

We first evaluate the performance of ReinforceWalk algorithm on these two tasks and compare it with other baseline methods. In Figure 5, we show the learning curves of the test accuracy for ReinforceWalk against other methods, and in Figure 6, we show the learning curves of the test MAP on the knowledge base completion (KBC) task. From the result, we can see that ReinforceWalk learns better policies faster than other baseline methods. To have more comprehensive comparison,

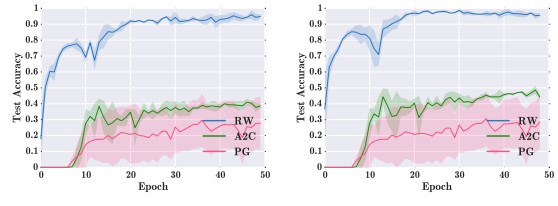

(a) Test Beam / Rollout = 128 (b) Test Beam / Rollout = 300

Figure 5: Test Accuracy on Three Glass Puzzle task. Higher is better. RW stands for ReinforceWalk. PG stands for policy gradient.

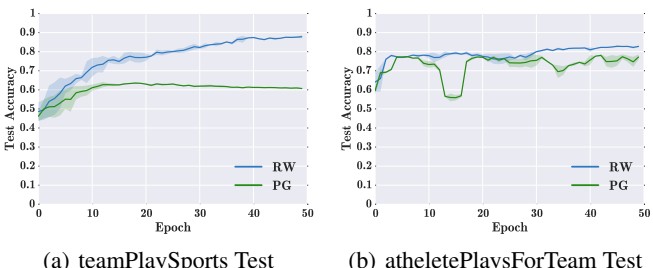

(a) teamPlaySports Test          (b) atheletePlaysForTeam Test

Figure 6: Test MAP on the KBC task (relations "teamPlaySport" and "atheletePlaysForTeam") Higher is better. RW stands for ReinforceWalk. PG stands for policy gradient.

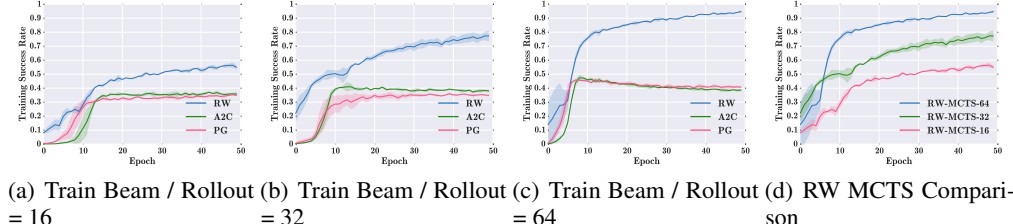

(a) Train Beam / Rollout (b) Train Beam / Rollout (c) Train Beam / Rollout (d) RW MCTS Compari-
= 16                      = 32                      = 64                      son

Figure 7: The training success rate (i.e., percentage of trajectories with positive reward during training) on Three Glass Puzzle task.

we report the final test MAP scores of ReinforceWalk on the knowledge base task (for all the 10 relations) in Table 1, and compare it to RL-based methods such as policy gradient (PG), advantage actor-critic (A2C), MINERVA, and DeepPath as well as non-RL baselines such as PRA, TransE and TransR. From the results in Table 1, we outperform previous work in most of the relations.

## A.3 ANALYSIS OF REINFORCEWALK

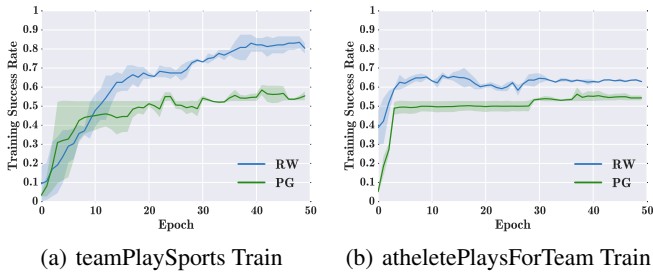

(a) teamPlaySports Train          (b) atheletePlaysForTeam Train

Figure 8: The training success rate (i.e., percentage of trajectories with positive reward during training) on the KBC task (relations "teamPlaySports" and "atheletePlaysForTeam").

Table 2: Three Glass Puzzle Test Accuracy, where "Beam" denotes beam search.

| Size | Test Accuracy (%) | | | |
|------|----------------|----------------|----------------|----------------|
| | PG (Beam) | A2C (Beam) | RW (Beam) | RW (MCTS) |
| 1 | $9.3 \pm 2.1$ | $9.7 \pm 4.0$ | $18.0 \pm 1.7$ | $18.0 \pm 1.7$ |
| 10 | $30.7 \pm 4.5$ | $22.3 \pm 1.5$ | $46.0 \pm 7.0$ | $63.3 \pm 5.0$ |
| 50 | $39.3 \pm 3.2$ | $34.3 \pm 3.1$ | $60.3 \pm 7.8$ | $84.3 \pm 3.1$ |
| 100 | $45.3 \pm 4.5$ | $39.3 \pm 2.3$ | $67.0 \pm 7.0$ | $90.7 \pm 2.5$ |
| 200 | $47.7 \pm 3.2$ | $46.3 \pm 3.2$ | $69.0 \pm 6.2$ | $95.0 \pm 2.6$ |
| 300 | $48.7 \pm 3.2$ | $46.0 \pm 1.0$ | $69.3 \pm 6.4$ | $96.3 \pm 1.5$ |
| 400 | $49.0 \pm 2.6$ | $46.3 \pm 1.5$ | $71.7 \pm 4.5$ | $99.0 \pm 1.0$ |

Table 3: BFS, DFS and ReinforceWalk on Three Glass Puzzle.

| Method | Average # Steps | Max # Steps |
|--------|-----------------|-------------|
| BFS | 264.7 | 1030 |
| DFS | 192.2 | 1453 |
| ReinforceWalk | 94.9 | 897 |

We now analyze the ReinforceWalk algorithm using different experiment results to understand the contributions of different components in ReinforceWalk. First, in Figures 7–8, we show the training success rate (i.e., percentage of trajectories with positive reward during training) on the Three Glass Puzzle task and the Knowledge Base Completion task. Compared to the policy gradient method (PG), and advantage actor-critic (A2C) methods, ReinforceWalk with MCTS is able to generate trajectories with more positive rewards, and this continues to improve as training progresses. This confirms our motivation of using MCTS to generate higher-quality trajectories to alleviate the sparse reward problem in graph walking. Furthermore, we also observe that with more MCTS simulations, the performance further improves, showing the importance of MCTS in training.

Second, to understand the importance of MCTS during testing, we compare the test accuracy across different algorithms with different beam search sizes and different MCTS rollouts during testing. The number of MCTS simulations for training ReinforceWalk is fixed to be 32. We repeat our experiments three times with different random seeds and report our experiments results in Table 2. We observe that ReinforceWalk with MCTS achieves the best test accuracy overall. In addition, with larger beam search sizes and MCTS rollouts, the test accuracy improves substantially. Furthermore, replacing the MCTS in ReinforceWalk by beam search at test time degrades the performance greatly, Therefore, MCTS is also very important for ReinforceWalk at test time.

As we mentioned earlier, the conventional graph traverse algorithms such as Breadth-First Search (BFS) and Depth-First Search (DFS) cannot be applied to graph walking problem as we do not know the ground truth target node at testing time. However, to understand how quickly ReinforceWalk with MCTS could find the correct target node, we compare it with the BFS and DFS in the following cheating setup for BFS and DFS. Specifically, we apply BFS and DFS to the test set of the Three Glass Puzzle task by disclosing the target node to them. In Table 3, we report the average traversal steps and maximum steps to reach the target node. The ReinforceWalk with MCTS algorithm is able to find the target node more efficiently compared to BFS and DFS. Finally, in Table 4, we present several examples of the knowledge graph traversal paths. Examples of Three Glass Puzzle traversal paths can be found in Tables 5 and 6.

## B ALGORITHM IMPLEMENTATION DETAILS

The detailed algorithm of ReinforceWalk is described in Algorithm 1.

### B.1 MCTS IMPLEMENTATION

In the MCTS implementation, we maintain a lookup table to record values $Q(s_t, a)$ and $N(s_t, a)$ for each visited state-action pair. The state $s_t$ in the graph walk problem contains all the information along the traversal path, and $n_t$ is the node at the current step $t$. We assign an index $i_a$ to each

Table 4: Examples of paths found by ReinforceWalk on the NELL-995 dataset. ReinforceWalk can learn to traverse multiple paths to reach the target node (example i) and learn to explore new paths when reach a dead end (example ii).

---

(i) **Search different paths after a dead end:** coach andrew brunette $\xrightarrow{\text{ATHLETEHOMESTADIUM}}$?

a) coach andrew brunette $\xrightarrow{\text{ATHLETEPLAYSFORTEAM}}$ sportsteam blackhawks $\xrightarrow{\text{ATLOCATION}}$ county chicago $\xrightarrow{\text{ATLOCATION}^{-1}}$

stadiumoreventvenue orchestra hall $\xrightarrow{\text{ATLOCATION}}$ county chicago $\xrightarrow{\text{STADIUMLOCATEDINCITY}^{-1}}$

stadiumoreventvenue united center $\xrightarrow{\text{TEAMHOMESTADIUM}^{-1}}$ sportsteam chicago bulls $\xrightarrow{\text{TEAMHOMESTADIUM}^{-1}}$ stadiumoreventvenue united center, ($V(s_T)$=0.383, False)

b) coach andrew brunette $\xrightarrow{\text{ATHLETEPLAYSFORTEAM}}$ sportsteam blackhawks $\xrightarrow{\text{ATHLETEPLAYSFORTEAM}^{-1}}$ athlete rich hill $\xrightarrow{\text{ATHLETEPLAYSFORTEAM}}$ sportsteam blackhawks $\xrightarrow{\text{TEAMHOMESTADIUM}}$ stadiumoreventvenue wrigley field, ($V(s_T)$=0.909, True)

ii) **Search over multiple paths to the target entity:** ceo evan williams $\xrightarrow{\text{PERSONLEADSORGANIZATION}}$?

a) ceo evan williams $\xrightarrow{\text{WORKSFOR}}$ company twitter, ($V(s_T)$=0.917, True)

b) ceo evan williams $\xrightarrow{\text{WORKSFOR}}$ company twitter $\xrightarrow{\text{ORGANIZATIONTERMINATEDPERSON}}$ ceo jack dorsey $\xrightarrow{\text{CEOOF}}$ company twitter, ($V(s_T)$=0.869, True)

iii) **worksfor** $\iff$ **organizationhiredperson**$^{-1}$: ceo garo h armen $\xrightarrow{\text{WORKSFOR}}$?

ceo garo h armen $\xrightarrow{\text{ORGANIZATIONHIREDPERSON}^{-1}}$ biotechcompany antigenics inc, ($V(s_T)$=0.935, True)

---

Table 5: ReinforceWalk Traversal Paths in Three Glass Puzzle, where "Index" stands for MCTS Path Index.

| Query | Index | ReinforceWalk Action Sequence | Estimated V |
|---|---|---|---|
| $(A, B, C, q)$ =(14, 45, 47, 15) | 0 | $(0, 0, 0) \rightarrow (14, 0, 0) \rightarrow (0, 14, 0) \rightarrow (14, 14, 0)$ $\rightarrow (0, 14, 14) \rightarrow (14, 14, 14) \rightarrow (0, 14, 28) \rightarrow (14, 14, 14) \rightarrow$ $(0, 14, 28) \rightarrow (14, 0, 28) \rightarrow (14, 45, 28) \rightarrow (14, 26, 47) \rightarrow$ END | 1.13e-6 |
| | 1 | $(0, 0, 0) \rightarrow (0, 0, 47) \rightarrow (14, 0, 33) \rightarrow (0, 14, 33) \rightarrow (14, 14, 19)$ $\rightarrow (0, 14, 19) \rightarrow (14, 0, 19) \rightarrow (14, 19, 0) \rightarrow (14, 19, 47) \rightarrow$ $(14, 45, 21) \rightarrow (14, 0, 21) \rightarrow (0, 14, 21) \rightarrow$ END | 6.31e-8 |
| | ... | ... | ... |
| | 14 | $(0, 0, 0) \rightarrow (0, 45, 0) \rightarrow (0, 0, 45) \rightarrow (14, 0, 31) \rightarrow (14, 45, 31) \rightarrow$ $(14, 29, 47) \rightarrow (14, 29, 0) \rightarrow (0, 29, 14) \rightarrow (14, 15, 14) \rightarrow$ END | 0.9999 |
| | ... | ... | ... |

Table 6: ReinforceWalk Traversal Paths in Three Glass Puzzle, where "Index" stands for MCTS Path Index.

| Query | Index | ReinforceWalk Action Sequence | Estimated V |
|---|---|---|---|
| $(A, B, C, q)$ =(11, 15, 30, 8) | 0 | $(0, 0, 0) \rightarrow (0, 0, 30) \rightarrow (11, 0, 30) \rightarrow (0, 11, 30) \rightarrow (11, 11, 19)$ $\rightarrow (0, 11, 30) \rightarrow (11, 11, 19) \rightarrow (0, 11, 30) \rightarrow (11, 0, 30)$ $\rightarrow (0, 11, 30) \rightarrow (11, 11, 30) \rightarrow (7, 15, 30) \rightarrow$ END | 2.13e-8 |
| | 1 | $(0, 0, 0) \rightarrow (0, 0, 15) \rightarrow (0, 0, 15) \rightarrow (0, 15, 15) \rightarrow (11, 4, 15)$ $\rightarrow (0, 4, 26) \rightarrow (4, 0, 26) \rightarrow (4, 15, 26) \rightarrow (11, 8, 26) \rightarrow$ END | 0.999 |
| | ... | ... | ... |
| | 4 | $(0, 0, 0) \rightarrow (11, 0, 0) \rightarrow (0, 0, 11) \rightarrow (11, 0, 11) \rightarrow (0, 11, 11)$ $\rightarrow (11, 11, 11) \rightarrow (0, 11, 22) \rightarrow (11, 11, 11) \rightarrow (7, 15, 11) \rightarrow$ $(7, 0, 11) \rightarrow (7, 15, 11) \rightarrow (7, 15, 30) \rightarrow$ END | 1.72e-8 |
| | ... | ... | ... |

candidate action $a$ from $n_t$, indicating $a$ is the $i_a$-th action of the node $n_t$. Thus the state $s_t$ can be encoded as a path string $P_{s_t} = (q, n_0, i_{a_0}, n_1, i_{a_1}, \ldots, n_t)$. We build a dictionary $\mathcal{D}$ using the path string as a key and we record $Q(s_t, a)$ and $N(s_t, a)$ as values in $\mathcal{D}$. In the backup stage, the $Q$ and $N$ values are updated for each state-action pair along with the traversal path in MCTS:

$$N(s_t, a) = N(s_t, a) + \gamma^{L-t}$$

$$Q(s_t, a) = Q(s_t, a) + \gamma^{L-t}V_\theta(s_L),$$

where $L$ is the length of the traversal path, and $\gamma$ is the reward discount.

---

**Algorithm 1** ReinforceWalk Training Algorithm

---
1: **Input:** Graph $\mathcal{G}$; Initial node $n_S$; Query $q$; Target node $n_T$; Maximum Path Length $T_{\max}$; MCTS Search Number $E$;
2: **for** episode $e$ in $[1..E]$ **do**
3:    Set current node $n_0 = n_S$; $q_0 = f_{\theta_q}(q, 0, 0, n_0)$
4:    **for** $t = 0 \ldots T_{\max}$ **do**
5:       Lookup from dictionary to obtain $Q(s_t, a)$ and $N(s_t, a)$
6:       Select the action $a_t$ with the maximum PUCT value:

$$a_t =$$
$$\arg\max_a \left\{ c \cdot \pi_\theta(a|s_t)^\beta \frac{\sqrt{\sum_{a'} N(s_t, a')}}{1 + N(s_t, a)} + \frac{Q(s_t, a)}{N(s_t, a)} \right\}$$

7:       Update $q_{t+1} = f_{\theta_q}(q_t, h_{A,t}, h_{a_t,t}, n_{t+1})$
8:       **if** $a_t$ is STOP **then**
9:          Compute estimated reward value $V_\theta(s_t)$
10:         Add generated path $p$ into a path list
11:         Backup along the path $p$ to update visit count $Q(s_t, a)$ and $N(s_t, a)$
12:         **Break**
13:      **end if**
14:   **end for**
15: **end for**
16: **for** each path $p$ in the path list **do**
17:    Set reward $r = 1$ if the end of the path $n_t = n_T$ otherwise $r = 0$
18:    Update the model parameters with Q-learning:

$$\theta \leftarrow \theta + \alpha \cdot \nabla_\theta Q_\theta(s_t, a_t) \times$$
$$\left( r(s_t, a_t) + \gamma \max_{a'} Q_\theta(s_{t+1}, a') - Q_\theta(s_t, a_t) \right)$$

   Update the Reward Estimation Network by minimizing the MSE loss:

$$\min_\theta \mathbb{E}(r(s_T, a_T) - V_\theta(s_T))^2$$

19: **end for**

---

## B.2 EXPERIMENT DETAILS

### B.2.1 KNOWLEDGE GRAPH LINK PREDICTION

The NELL-995 knowledge dataset contains $75,492$ unique entities and $400$ relations. We set the entity embedding dimension to $4$ and relation embedding dimension to $64$. The maximum length of graph walking path is $8$, which indicates only the STOP action can be taken in the last step. After the STOP action being taken, the system evaluates the action sequence and assigns a reward $r = 1$ if the agent reaches the target node, otherwise $r = 0$. The initial query $q$ is the concatenation of the entity embedding vector and the relation embedding vector. The $f_{\theta_S}$ and $f_{\theta_A}$ functions are modeled by two different DNNs with the same architecture: two fully-connected layers with 64 hidden dimensions and the ReLU activation function. $f_{\theta_v}$ is a two fully-connected layers with 16 hidden dimensions, where the first hidden layer is with Tanh activation function and the output layer with linear activation function. $f_{\theta_q}$ is a modeled by a GRU with hidden size 64. The hyperparameters in PUCT is set as $c = 2$ and $\beta = 0.5$. We rollout 32 MCTS paths in the training and prediction. We use ADAM optimization algorithm for model training with learning rate 0.0001 and we set the mini-batch size as 8.

### B.2.2 THREE GLASS PUZZLE

In the experiments of Three Glass Puzzle, we randomly draw four integers from $[1, 50)$ to represent the capacity of $A, B, C$ and the desired volume $q$, respectively. We restrict each puzzle to be

$A \leq B \leq C$ and $q < C$ to avoid data duplication. We clean up the puzzles if there is no solution to them. Finally, we keep 600 unique puzzles as the experimental dataset, where 500 puzzles are used for training and the other 100 are used to test model's generalization capability on unseen test set.

Let $a, b, c$ be the current status of each container, we define the puzzle status at step $t$ as $n_t = [I_A^T, I_B^T, I_C^T, I_a^T, I_b^T, I_c^T]^T$, where $I_x$ is the one-shot representation to encode the value of $x$. Given $A, B, C, a, b$ and $c$ are all smaller than $50$ in the experiment, the dimension of $n_t$ is 300. The initial query $q$ is obtained by $q = E[q]$, where $E$ is a query embedding lookup table and $E[x]$ indicates the $x$-th column. The query embedding dimension is set as $64$.

In the three glass puzzle, there are 13 actions in total: fill one container to its capacity, empty one container to empty, pour one container to the other container, and a STOP action to terminate the game. We set the maximum length of action sequence as 12, which indicates only the STOP action can be taken in the last step. After the STOP action being taken, the system evaluates the action sequence and assigns a reward $r = 1$ if the final status is a success, otherwise $r = 0$.

The $f_{\theta_S}$ and $f_{\theta_A}$ functions are modeled by two different DNNs with the same architecture: two fully-connected layers with 32 hidden dimensions and ReLU activation function. $f_{\theta_v}$ is a two fully-connected layers with 16 hidden dimensions, where the first hidden layer is with ReLU activation function and the output layer is with linear activation function. $f_{\theta_q}$ is a modeled by a GRU with hidden size 64. The hyperparameters in PUCT is set as $c = 0.5$ and $\beta = 0.2$. We use ADAM optimization algorithm with learning rate $0.0005$ during training and we set the mini-batch size as $8$.

Table 7: A List of Actions for Each Container in the Three Glass Puzzle. The agent can also determine to take the STOP action to terminate the game.

| Empty A | Fill A | Pour A to B | Pour A to C |
|---------|--------|-------------|-------------|
| Empty B | Fill B | Pour B to A | Pour B to C |
| Empty C | Fill C | Pour C to A | Pour C to B |

