# OpenReview forum: "ReinforceWalk: Learning to Walk in Graph with Monte Carlo Tree Search"
_ICLR.cc/2018/Workshop — Accept_

### Official Review · AnonReviewer1 · 2018-03-11
**Uses MCTS + RL for knowledge base completion**

**Rating:** 6
**Confidence:** 4

**Review:**

This paper uses MCTS + RL for answering queries on a knowledge base graph. The contribution of this paper is to do planning using MCTS instead of policy gradient. This indeed is important in the sparse reward setting and leads to better peformance.
1. Given that this approach is so close to Minerva, please explicitly contrast against it at the beginning of the paper. The basic explanation is hurried and not very clear.
2. In the line, "we use the following formula to calculate the score for each unique candidate n. .." Please justify this. Isn't there a more principled way to address this issue?

---

### Official Review · AnonReviewer3 · 2018-03-11
**Cool application of PG + MCTS to KB reasoning, potentially not novel enough**

**Rating:** 5
**Confidence:** 3

**Review:**

This paper introduces a method which combines Policy Gradient (or other model-free techniques such as A2C) with model-based reasoning powered by MCTS. Since the RL dynamics are known and deterministic MCTS naturally improves performance (much like in e.g. AlphaGo). The idea of using MCTS + deep RL is far from new, so the originality of this work is, in my opinion, quite weak. The only novel idea is how to deal with cycles in the knowledge graph traversal, and the fix is quite trivial.

The application and good empirical results on tough KB reasoning datasets is quite good however.

One important note: I disagree with the authors that this is a "partially observable" MDP, just because you don't base your decision on the current node doesn't mean the other nodes are unobserved -- they are observed just not included in your state vector.

---

### Public Comment · ~Rajarshi_Das1 · 2018-03-04
**MINERVA results**

Hi!,

Disclosure: I am one of the authors of MINERVA.

This is a nice idea and it is good to see it performs well!.

 I wanted to comment on the performance of MINERVA^{a} in Table 1. We also tried MINERVA in the same reported setting (training per-relation models) and we got significantly different results. We have updated our paper (https://openreview.net/pdf?id=By1ZYf-0b) with the latest scores (Table 5, column 3). These results are very close to what RW gets. We would be happy to help figure out what is the source of the discrepancy.

Rajarshi

---

> ### Public Comment · ~Po-Sen_Huang1 · 2018-03-05
> **Reply to "MINERVA results"**
>
> Thanks for the update! We got the MINERVA baselines in our paper by running the source code (and some details shared by your colleagues) you released earlier. We are now re-running experiments based on the codes you recently updated, and we will revise our report of MINERVA once the new results are confirmed.

---

### Public Comment · ~Sachin_Rajoria2 · 2018-03-05
**Please help and explain a new comer in the field**

I am trying to learn about the field of reasoning over KBs. I have read papers using methods like classical PRA, tensor factorization approaches, logic-based approaches, etc. I recently read the MINERVA paper from the main ICLR 2018 conference and now ReinforceWalk. According to my understanding, the MINERVA paper took a novel approach and framed the QA on KB as RL on a graph constructed from KB, and the agent was trained to navigate the environment using REINFORCE algorithm. Please correct me if I am wrong, but the problem formulation in terms of RL in ReinforceWalk seems to be exactly same as MINERVA, and the training was carried out using standard MCTS instead of REINFORCE. Can you please explain if I am missing anything? In case the difference between MINERVA and ReinforceWalk is only in the training algorithm, it would be more convincing if there are experimental evaluations over many datasets to showcase advantage/disadvantage of one training algorithm over the other or to showcase in which setting MCTS is better and in which REINFORCE is. Right now I am not clear which one to use for my application. I would be grateful for your response and would be very helpful in my learning of the field.

---

> ### Public Comment · ~yelong_shen1 · 2018-03-05
> **Reply to "Please help and explain a new comer in the field"**
>
> Hi Sachin,  "DeepPath" and "MINERVA" are recently two approaches, which utilize RL algorithms for KB Reasoning task.  ReinforceWalk uses the same problem formulation as in "MINERVA". With slightly different model architectures of "MINERVA" and "ReinforceWalk", ReinforceWalk tries to target sparse reward issue, and incorporate planning mechanism in the graph walk task.  We compared "RW" and "PG" ( policy gradient : which is the training algorithm in MINERVA) in our experiments to show the advantage of mcts and planning. We also compare RW and PG in several datasets in our arxiv preprint.  It could be great if you could let us know which other datasets you are interested to give a comparison with "RW" and "MINERVA".  Meanwhile, we are going to release the source code in the next month, you could also be free to try it out.

---

> > ### Public Comment · ~Sachin_Rajoria2 · 2018-03-06
> > **Thank you**
> >
> > Hi Yelong, Thank you so much for the detailed response and confirming my belief that only difference MINERVA and ReinforceWalk is the inference algorithm. I guess there is no mathematically guaranteed superiority of one inference technique over the other between REINFORCE based policy gradient inference in MINERVA and standard MCTS based inference in ReinforceWalk. The release of the code would be super helpful, I can try out both for my application and maybe MCTS based approach works out!

---

### Decision · Program_Chairs · 2018-03-20
**ICLR 2018 Workshop Acceptance Decision**

**Decision:**

Accept

**Comment:**

Congratulations, your paper was accepted to the ICLR workshop.